# Mass spectrometry imaging of SOD1 protein-metal complexes in SOD1G93A transgenic mice implicates demetalation with pathology

Oliver J. Hale [1], Tyler R. Wells [2], Richard J. Mead [2,3] ✉ & Helen J. Cooper [1] ✉

Amyotrophic lateral sclerosis (ALS) is characterized by degeneration of motor neurons in the central nervous system (CNS). Mutations in the metalloenzyme SOD1 are associated with inherited forms of ALS and cause a toxic gain of function thought to be mediated by dimer destabilization and misfolding. SOD1 binds two Cu and two Zn ions in its homodimeric form. We have applied native ambient mass spectrometry imaging to visualize the spatial distributions of intact metal-bound SOD1[G93A] complexes in SOD1[G93A] transgenic mouse spinal cord and brain sections and evaluated them against disease pathology. The molecular specificity of our approach reveals that metal-deficient SOD1[G93A] species are abundant in CNS structures correlating with ALS pathology whereas fully metalated SOD1[G93A] species are homogenously distributed. Monomer abundance did not correlate with pathology. We also show that the dimer-destabilizing post-translational modification, glutathionylation, has limited influence on the spatial distribution of SOD1 dimers.

Mutations in the gene coding the metalloenzyme superoxide dismutase 1 (SOD1) are responsible for ~20% of familial amyotrophic lateral sclerosis (fALS) cases, an inherited disease characterized by degeneration of neurons in motor-associated regions of the central nervous system (CNS)[1]. Under physiological conditions, the human wild-type SOD1 (hSOD1[wt]) matures to form a non-covalently bound homodimer incorporating one zinc ion, one copper ion and one intramolecular disulfide bond per subunit. The canonical view is that mutations in *SOD1* destabilize the native structure of the protein, resulting in monomerization and aggregation of the protein. This pathway results in a toxic gain of function that causes degeneration of motor neurons[1,2]. Demetalation[3–7], absence of the intramolecular disulfide bond[8], loss of secondary structure[9], monomerization[10], oligomerization[7,9,11], and oxidative post-translational modification[12,13] have all been associated with SOD1 misfolding. Some studies have also suggested that misfolding of

wild-type SOD1 extends to cases of sporadic ALS[14–17], although this remains controversial[18,19].

Here, we investigated the distribution of SOD1 complexes in sections of spinal cord and brain from the well-established transgenic hSOD1[G93A] mouse model of ALS[20] and the transgenic hSOD[wt] control using a recently developed native ambient mass spectrometry imaging (MSI) methodology. Native ambient MSI is a label-free molecular imaging technique with the unique capability to identify and map the distribution of endogenous protein complexes within tissue sections, including metal-bound proteins and membrane proteins[21–24]. Unlike mass spectrometry-based elemental imaging techniques, which provide information on total metal ion distribution within tissue but lack specificity for protein–metal binding[25,26], and matrix-assisted laser desorption/ionization (MALDI) MSI, in which protein distributions can be visualized but any information on metal binding is lost due to the denaturing sample preparation conditions[27], native ambient MSI

[1]School of Biosciences, University of Birmingham, Birmingham, UK. [2]Sheffield Institute for Translational Neuroscience, University of Sheffield, Sheffield, UK. [3]Neuroscience Institute, University of Sheffield, Western Bank, Sheffield, UK. ✉e-mail: r.j.mead@sheffield.ac.uk; h.j.cooper@bham.ac.uk

involves direct analysis of the intact protein−metal complex. Native ambient MSI can distinguish signals specific to different numbers of non-covalently bound metal ions and post-translationally modified forms of proteins (proteoforms). Fresh-frozen tissue is analyzed intact, i.e., there is no requirement for homogenization, nor is there a requirement for the development of specific antibodies or tags. The workflow is described in Fig. 1, with amino acid sequences for wild-type mouse SOD1 (mSOD1[wt]), wild-type human SOD1 (hSOD1[wt]), and human G93A SOD1 (hSOD1[G93A]) variant given in Supplementary Fig. S1 and details of the samples given in Supplementary Table S1, Supplementary Information. Our observations indicate that localized abundance of metal-deficient states of hSOD1[G93A] is a key factor in the degeneration of motor neurons in the spinal cord and brain.

## Results

### Identification and characterization of protein complexes

Characterization of proteins and protein complexes detected in the CNS tissues was performed by top-down mass spectrometry after sampling with nanospray-desorption electrospray ionization (nano-DESI[28]) under native conditions[21], or liquid extraction surface analysis (LESA[29]) under denaturing conditions[30]. For brevity, details of characterization experiments are included in Supplementary Note 1, Supplementary Figs. S2–S14, and Supplementary Tables S2–S6. In summary, hSOD1[G93A] and hSOD1[wt] were detected as dimers binding 2, 3 (metal-deficient) and 4 (holo) metal ions, and as monomers binding 1 (metal-deficient) or 2 (holo) metal ions.

### MS imaging of protein−metal complexes

**Spinal cord.** Motor neuron degeneration in the lumbar region of the spinal cord is well-established in the hSOD1[G93A] mouse model[20,31]. Representative spatial analyses of protein complexes by native ambient MSI in lumbar cord sections for each genotype (hSOD1[wt] and hSOD1[G93A]) are shown in Fig. 2. Ion images of each protein charge state contributing to the images in Fig. 2, and biological replicates, are included in Supplementary Figs. S15 (hSOD1[G93A]), S16 (hSOD1[wt]), and

S17 (monomers), Supplementary Information. We used the protein complex [Arf1 + GDP] (identified in the previous work[24]), a molecular marker for the grey matter, to define the section outline (Fig. 2b−i) and assist with image interpretation.

SOD1 complexes were detected in multiple metal-bound states; hSOD1[wt] was predominantly detected in the holo-form, binding a total of 4 metal ions in the dimeric complex (Fig. 2e, 2 metal ions for the monomer; Fig. 2g). Conversely, hSOD1[G93A] was detected in metal-deficient forms, binding only 2 (Fig. 2j) or 3 metal ions (Fig. 2k) in each dimer (1 metal ion for the monomeric form, Fig. 2m), in addition to the holo-form. Metal-deficient dimers and monomers of hSOD1[G93A] featured the same spatial distributions. We found that metal-deficient dimeric and metal-deficient monomeric hSOD1[G93A] complexes were significantly more abundant in the ventral horn (Fig. 2o, dimers; $P < 0.01$, monomers; $P < 0.001$) than in the dorsal horn, correlating with fALS pathology. Fully metalated (i.e., dimers with 4 metal ions, monomers with 2 metal ions) hSOD1[G93A] complexes were ubiquitous with no significant difference in abundance between ventral and dorsal horns, and high-resolution native ambient MS imaging showed lower abundance in grey matter versus white matter. This observation supports the hypothesis that the metal-deficient state is key to the disease pathology, rather than monomerization, since fully metalated monomers were also abundant in hSOD1[G93A] dorsal horn (Fig. 2n, o). Metal-deficient complexes were not detected above baseline chemical signal in the hSOD1[wt] spinal cord tissue. No significant difference in abundance was observed between hSOD1[wt] complexes detected in the ventral horn and dorsal horn. Overall, signal intensity was greater for hSOD1[G93A] than for hSOD1[wt], reflective of the higher expression level of SOD1 in the hSOD1[G93A] model but not linked with disease pathology[20].

Examination of formalin-fixed paraffin-embedded (FFPE) tissue sections from hSOD1[G93A] and mSOD1[wt] littermates (Supplementary Fig. S18, Supplementary Information) confirmed the reduced numbers of motor neurons in the ventral horn at 120 days of age in the hSOD1[G93A] model (Fig. 2p), correlating with the elevated abundance of metal-deficient hSOD1[G93A] complexes observed in the mass

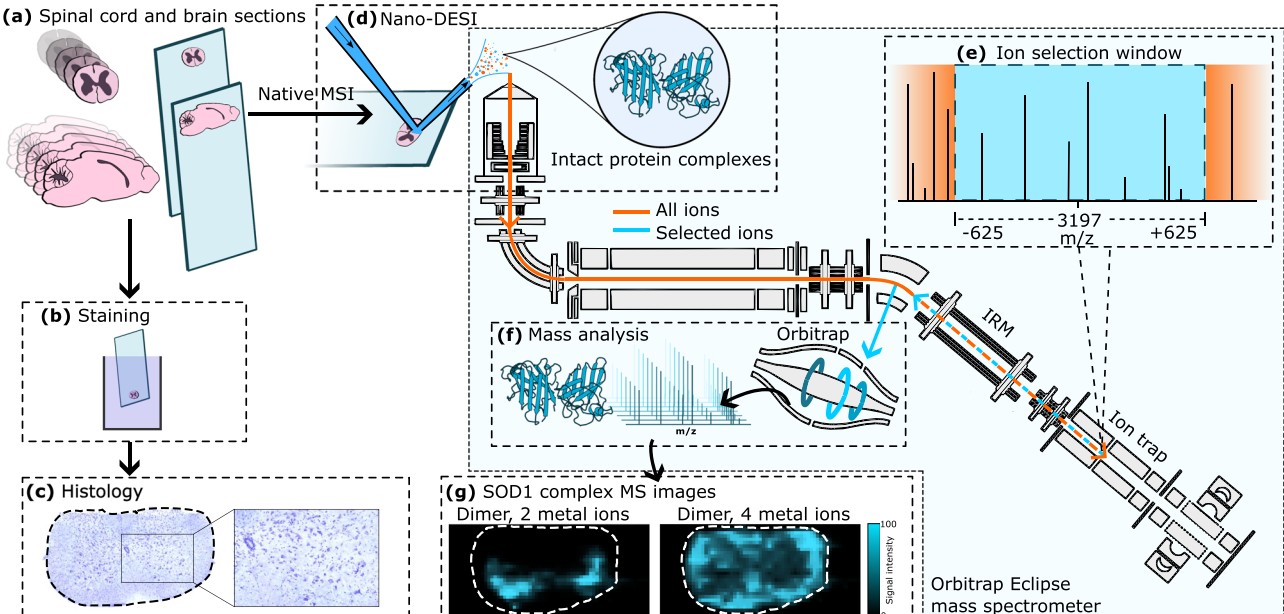

**Fig. 1 | The workflow for the analysis of the spatial distribution of hSOD1[G93A] complexes in mouse CNS tissues by native ambient MSI. a** Serial spinal cord sections were taken in the coronal plane and brain sections in the sagittal plane. **b** Nissl staining for visualization of neurons (some sections were stained post-native ambient MSI). **c** Histology to determine motor neuron abundance. **d** Nano-DESI sampling and ionization of protein complexes directly from a tissue section. **e** Protein ions were transmitted into an Orbitrap Eclipse mass spectrometer and transferred to the linear ion trap where a wide $m/z$ selection window was applied, e.g., $m/z$ 3197 ± 625. The orange line represents the ion beam before $m/z$ selection in the ion trap. The cyan line represents ions selected by the ion trap. **f** The selected ions were transmitted to the orbitrap mass analyzer for $m/z$ measurement. **g** Protein images were generated and compared with histological data.

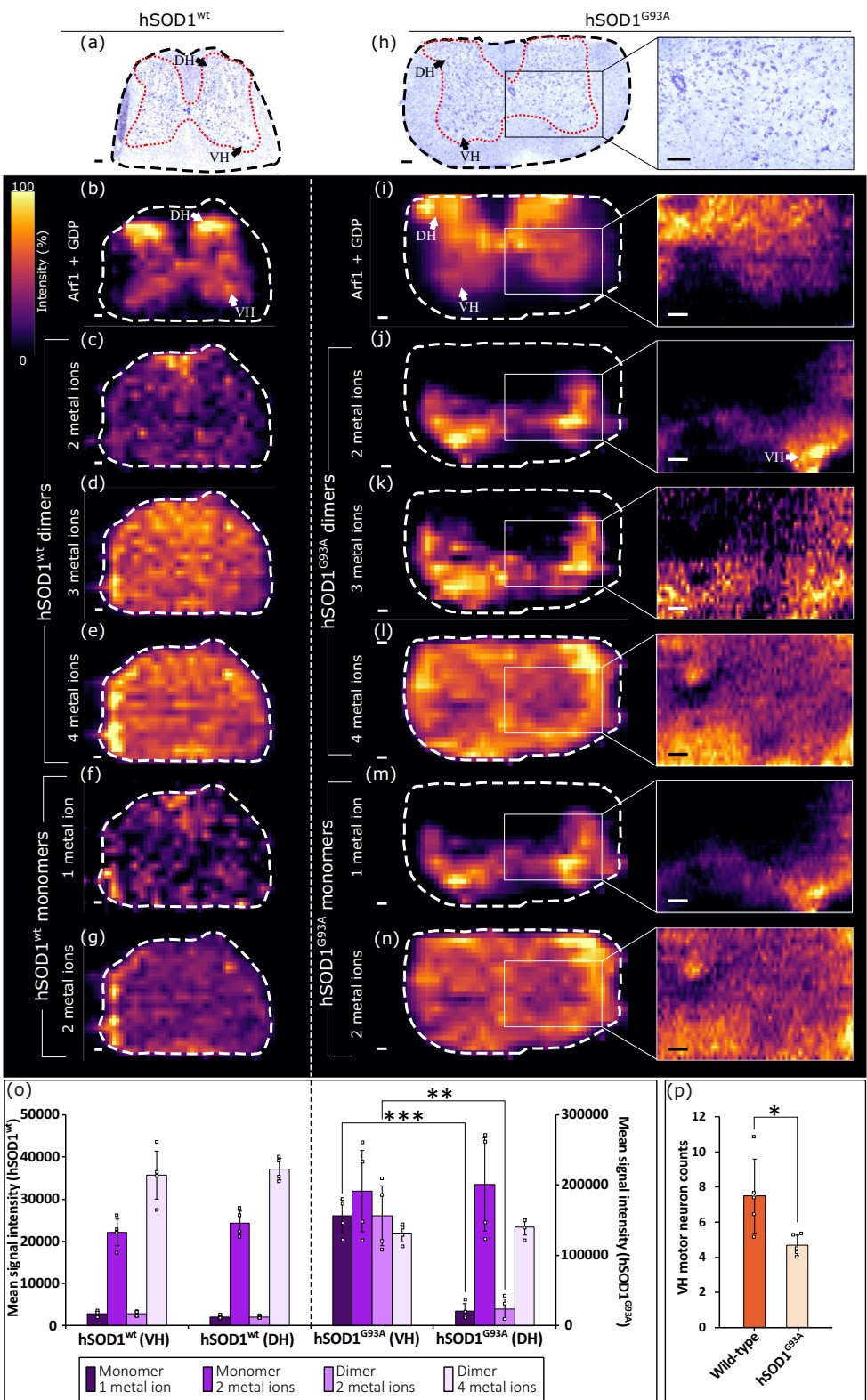

**Fig. 2 | Histology and MS images of transgenic mouse lumbar spinal cord. hSOD1$^{wt}$. a** Fresh-frozen, Nissl-stained serial tissue section (grey column indicated by the red dashed line) of spinal cord hWT-SC1 (representative of 4 total sections). **b** The ion image for the Arf1+GDP complex depicts the grey matter column. Ion images for hSOD1$^{wt}$ dimers with (**c**) two metal ions; **d** three metal ions, **e** four metal ions, and monomers with (**f**) one metal ion; **g** two metal ions. **hSOD1$^{G93A}$: h** Fresh-frozen, Nissl-stained serial tissue section (grey column indicated by the red dashed line) and expanded view of a ventral horn of spinal cord G93A-SC1 (representative of 6 total sections). Ion images for (**i**) [Arf1+GDP] complex; hSOD1$^{G93A}$ dimers with (**j**) two metal ions; **k** three metal ions; **l** four metal ions, and monomers with (**m**) one

metal ion; **n** two metal ions. White boxes show high-resolution MSI for the area indicated. **o** Evaluation of relative abundances of SOD1 complexes in DH and VH for each genotype. Data expressed as mean signal from 4 VH and DH +/− SD. **p** Average motor neuron counts (+/− SD) in mSOD1$^{wt}$ (*N* = 5) and hSOD1$^{G93A}$ (*N* = 5) mice in the VH of lumbar cord sections (mSOD1$^{wt}$ *N* = 28, hSOD1$^{G93A}$ *N* = 36) with at least 50 µm between sections. Students' two-sided *T* test; *$P$ < 0.05 (actual $P$ = 0.021), **$P$ < 0.01 (actual $P$ = 2.19 × 10$^{-3}$), ***$P$ < 0.001 (actual $P$ = 9.07 × 10$^{-5}$). DH dorsal horn, VH ventral horn. Scale bars; 100 µm (**a**–**n**). MS images have one order of linear interpolation applied and a linear intensity scale. Source data are provided as a Source Data file.

spectrometry imaging experiments. Significantly lower motor neuron counts were recorded in the hSOD1[G93A] spinal cord tissue ($P < 0.05$) with respect to mSOD1[wt]. (Note, the mSOD1[wt] tissue serves as a proxy for hSOD1[wt] tissue, which was only available fresh-frozen and was unsuitable for neuron counting. Mice of both wild-type genotype do not exhibit motor neuron degeneration[20]). We conclude that there is a critical relationship between hSOD1[G93A] metal deficiency and the degeneration of motor neurons in the spinal cord.

**Brain.** Histology and native ambient MS images were obtained from hSOD1[wt] and hSOD1[G93A] sagittal brain sections. Figure 3 shows images for a representative analysis of each genotype. Ion images for individual protein charge states are shown in Supplementary Figs. S19, 20 (hSOD1[wt]) and S21, 22 (hSOD1[G93A]), Supplementary Information. Composite images for biological replicates are shown in Supplementary Figs. S23 and S24. The zinc-bound carbonic anhydrase 2 complex [CAH2 + Zn[2+]] was used as a biomolecular marker for white matter, e.g., corpus callosum (Fig. 3b–i)[24].

In wild-type mice, the majority of hSOD1[wt] was detected in the holo-form, i.e., fully metalated. Lower abundance metal-deficient (2 and 3 metal ions) hSOD1[wt] dimers were also detected localized to the hippocampal formation (Fig. 3c, d), possibly within pyramidal cells[32]. Holo-hSOD1[wt] (Fig. 3e) and holo-hSOD1[G93A] (Fig. 3l) dimers were abundant and ubiquitous throughout the sections. Holo-SOD1 is highly abundant in cerebrospinal fluid (CSF)[33], which correlates with greater signal intensity in ventricular regions and the cerebral aqueduct. Metal-deficient hSOD1[G93A] dimers showed a notably different spatial distribution to the wild-type, with the greatest signal intensity in cranial nuclei of the brainstem and no notable abundance in the hippocampus (Fig. 3j, k). Structures related to facial motor functions, such as swallowing (e.g., hypoglossal nucleus, XII) and mastication (e.g., trigeminal nucleus, TN; facial nucleus, VII), contain degenerating motor neurons in ALS[34]. Metal-deficient hSOD1[G93A] was of low intensity in ventricles suggesting low relative abundance in CSF and that metal-deficient hSOD1[G93A] complexes are formed in situ owing to the local cellular environment.

Monomeric hSOD1 was detected in both hSOD1[wt] and hSOD1[G93A] brains. As mentioned above, monomerization is believed to be a key event in toxic gain of function, i.e., misfolded SOD1 results in the formation of monomers that subsequently aggregate into insoluble inclusion bodies. Ion images for monomeric hSOD1[wt] (Fig. 3f, g) and hSOD1[G93A] (Fig. 3m, n) showed similar spatial distributions to the dimers of equivalent metal-binding state, that is, the distribution of holo-monomers (2 metal ions) matched that of holo-dimers (4 metal ions), and metal-deficient monomers (1 metal ion) matched that of metal-deficient dimers (2 or 3 metal ions). (For monomers in biological replicates, see Supplementary Fig. S25, Supplementary Information).

The region of the brainstem featuring the hypoglossal nucleus (XII) has high signal intensity for hSOD1[G93A] dimers binding 2 metal ions. The intensity differs significantly ($P < 0.05$) to the signal intensity for the same complex in the frontal cortex (Fig. 3o), a region not associated with ALS pathology, and correlates with significantly ($P < 0.01$) lower numbers of motor neurons observed in Nissl-stained XII compared to wild-type mouse (Fig. 3p and Supplementary Fig. S26, Supplementary Information). In both regions in the wild-type genotype and in the hSOD1[G93A] cortex, fully metalated complexes are most abundant and feature similar relative abundances. These results suggest that metal-binding state, rather than monomerization, is the key driver of pathological change. That is, the results in Fig. 3 imply a link between the localization of metal-deficient hSOD1[G93A] and degenerating motor nuclei in the brainstem, and correlate with the results from the spinal cord.

## Evaluation of metal binding and PTMs
In all cases, hSOD1 (wild-type and hSOD1[G93A]; monomeric, dimeric, fully metalated, metal-deficient) was detected without the initiator

methionine and was acetylated at the N-terminus. Further modification by larger covalent PTMs is known to destabilize SOD1 structure and decrease its activity[12,13,35]. Glutathionylation of hSOD1 (an additional ~305 Da per glutathionyl group) at Cys-111 has been proposed to contribute to ALS pathogenesis by destabilization of the dimer interface and increasing of the dimer dissociation constant[12,13]. We identified N-acetyl, S-glutathionyl (GS-) hSOD1[G93A] in tissue by intact mass measurement and top-down MS. Dimers containing 1 (1GS-dimer) and 2 (2GS-dimer) S-glutathionylated hSOD1[G93A] subunits were imaged at moderate resolution for full brain sections (composite images: Supplementary Figs. S24 and S27. Individual protein charge states: Supplementary Fig. S28, Supplementary Information). Higher resolution native ambient MSI of the brainstem revealed that metal-deficient hSOD1[G93A] homodimers and metal-deficient 1GS-dimers (hSOD1[G93A]/GS- hSOD1[G93A]), each bound to 2 metal ions, exhibited the same localization, i.e., to cranial nuclei (Fig. 4a, b and Supplementary Fig. S22 Supplementary Information). The equivalent holo-dimers (i.e., 4 metal ions) were ubiquitous, with the greatest abundance in the fourth ventricle (Fig. 4c, d).

High-resolution native ambient MSI also enabled analysis of tissue structures of the cerebellum (Fig. 4e–j and Supplementary Fig. S29, Supplementary Information). The involvement of the cerebellum in ALS progression is not commonly the focus of ALS pathology, despite evidence of its importance[36–38]. Metal-deficient hSOD1[G93A] homodimers and hSOD1[G93A]/GS-hSOD1[G93A] 1GS-dimers were observed in cerebellar grey matter corresponding to the deep cerebellar nuclei (DCN)[39] and cerebellar peduncles (CBP) (Fig. 4g, h). Holo-homodimers were ubiquitous (Fig. 4i), but holo-1GS-dimers were most abundant in the molecular layer of the cerebellar cortex (Fig. 4j). This observation suggests an instance where spatial distribution of hSOD1[G93A] dimers is influenced by covalent PTMs, but further investigation is required. While PTMs may play a role in SOD1 dimer destabilization and fALS progression, MSI indicates that the number of bound metal ions is the key differentiator in hSOD1[G93A] complex localization in vivo.

## Implications of hSOD1 metal-binding state
Wild-type-like SOD1 mutants such as hSOD1[G93A] possess a mutation outside of metal-binding regions of the protein structure and enzymatic function is retained at wild-type levels[3,20]. Demetalation is known to occur for hSOD1[G93A], and Cu-mediating therapeutics have been shown to improve the longevity of transgenic mice[40–42]. Tokuda et al. developed an antibody (anti-apoSOD) specific for Cu-deficient hSOD1[4]. Extracts of homogenized tissue from the brainstem, cerebellum, and spinal cords of hSOD1[G93A] mice were analyzed by sandwich ELISA, and signals were observed in the spinal cord extracts at the presymptomatic stage but not at end stage. ELISA signals were not observed for the brainstem or cerebellum samples. This result contrasts with our findings which clearly show metal-deficient hSOD1[G93A] in regions of the spinal cord, the brainstem, and cerebellum. This discrepancy is likely because, in the ELISA experiments, the entire brainstem (~40 mg) was homogenized: Any spatial information was lost and any differential signal was averaged in the homogenate. Moreover, immunostaining of sections of the spinal cord from wild-type and hSOD1[G93A] mice at various ages with anti-apoSOD was not successful, with significant background staining and high irreproducibility[4]. Our results demonstrate successful imaging of metal-deficient hSOD1[G93A] in tissue sections.

The MS images presented here indicate that the accumulation of metal-deficient hSOD1[G93A] complexes in motor-associated CNS regions is related to motor neuron degeneration. The minimum number of metal ions incorporated into hSOD1[G93A] dimers observed in our study was two. Presumably, Zn-binding sites were occupied as studies have shown Zn incorporation to be structurally critical, enabling interaction with the copper chaperone, CCS[43,44]. CCS is required for the maturation of SOD1 from a disulfide-reduced, copper-free state to one with an

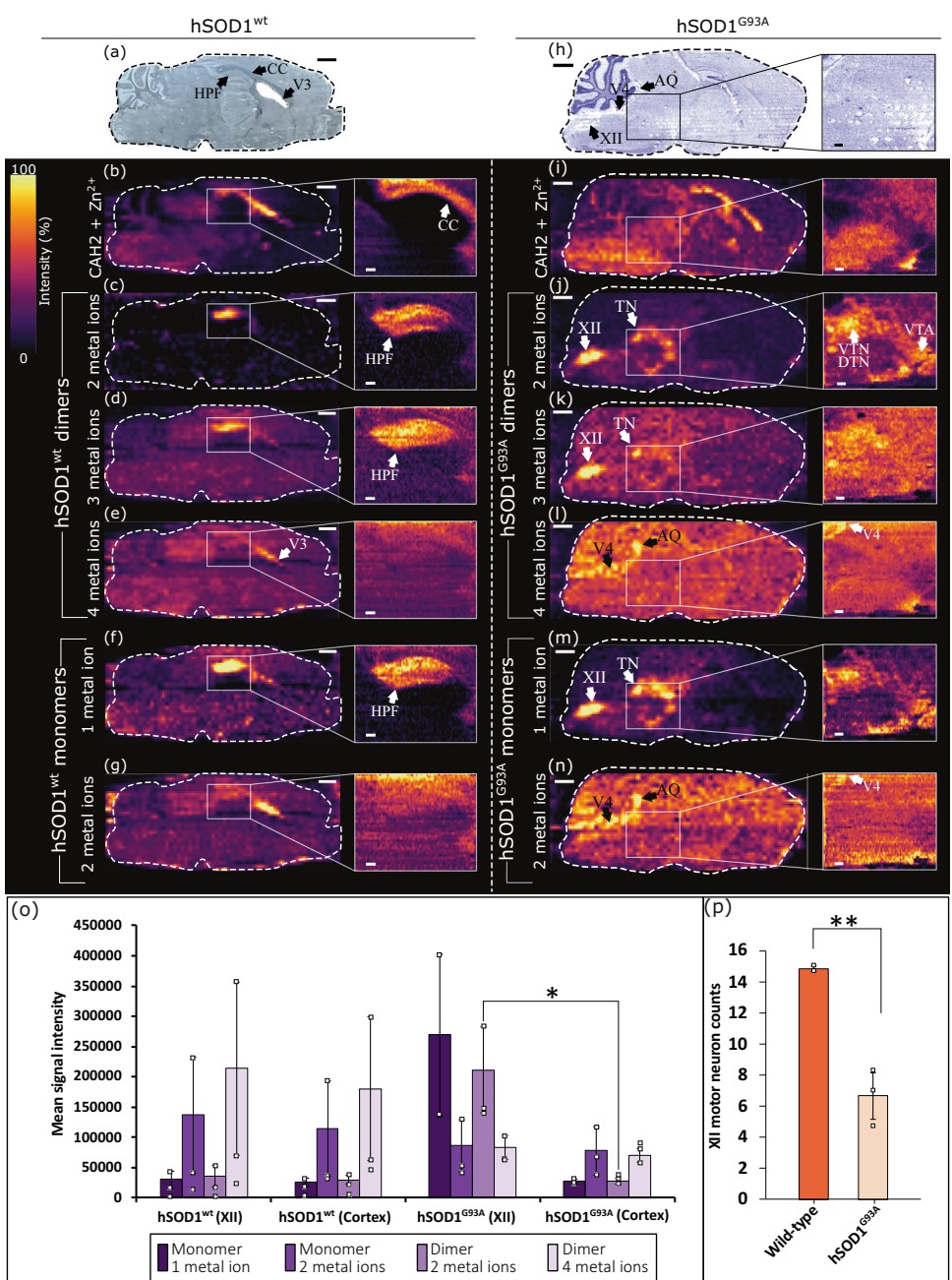

**Fig. 3 | Histology and MS images of transgenic mouse brains. a** optical scan of a section of brain hWT-B2. MS images for (**b**) carbonic anhydrase (CAH) 2 + Zn$^{2+}$, intact hSOD1$^{wt}$ dimers, (**c**) with two metal ions; **d** three metal ions; **e** four metal ions, and monomers with (**f**) one metal ion; **g** two metal ions. Selected regions show high-resolution MSI for the tissue around the hippocampal formation. **h** Nissl-stained (post-MSI) sections of brain G93A-B1. Full section MS images for (**i**) CAH2 + Zn$^{2+}$, intact hSOD1$^{G93A}$ dimers with (**j**) two metal ions; **k** three metal ions; **l** four metal ions, and monomers with (**m**) one metal ion; **n** two metal ions. White boxes show high-resolution MSI. **o** Relative signal intensity of SOD1 complexes in hypoglossal nucleus (XII) and frontal cortex for each genotype (hSOD1$^{wt}$ $N = 3$, hSOD1$^{G93A}$ $N = 3$). Data expressed as mean signal intensity from three sections for each genotype +/−

SD. **p** Average motor neuron counts (+/− SD) in XII from mSOD1$^{wt}$ ($N = 2$, total sections = 12) and hSOD1$^{G93A}$ ($N = 3$, total sections = 15) mice. Students' two-sided $T$ test; *$P < 0.05$ (actual $P = 0.027$), **$P < 0.01$ (actual $P = 9.45 \times 10^{-3}$). HPF hippocampal formation, CC corpus callosum, V3 third ventricle, V4 fourth ventricle, AQ cerebral aqueduct, XII hypoglossal nucleus, TN tegmental nuclei, VTN ventral tegmental nucleus, DTN dorsal tegmental nucleus, VTA ventral tegmental area. MS images have one order of linear interpolation applied and a linear intensity scale. Scale bars: Full brain images; 1000 μm, high-resolution MSI; 200 μm, Nissl-stain expanded region; 50 μm. High-resolution MSI was performed on a separate serial section. Source data are provided as a Source Data file.

intramolecular disulfide bond and Cu ions[43–45].The abundance of Zn-bound dimers in the motor-associated regions supports the assertion that maturation is hindered at the disulfide formation/Cu ion incorporation step. An increased oxidative environment in the cytosol of cells in the affected motor regions could cause premature intramolecular disulfide formation prior to interaction with CCS. CCS cannot interact with Zn-bound SOD1 dimers with a preformed disulfide bond,

thus preventing maturation[44]. As described in Supplementary Fig. S10, Supplementary Information, some proportion of the metal-deficient hSOD1$^{G93A}$ dimers (2 metal ions) featured intact disulfide bonds. An insufficient level of Cu for SOD1 maturation has been reported in bulk CNS tissues of the hSOD1$^{G37R}$ mouse model, whereas Zn levels were elevated in line with protein overexpression[46]. Evidence of Cu dyshomeostatis and wild-type SOD1 dysfunction has also been observed in

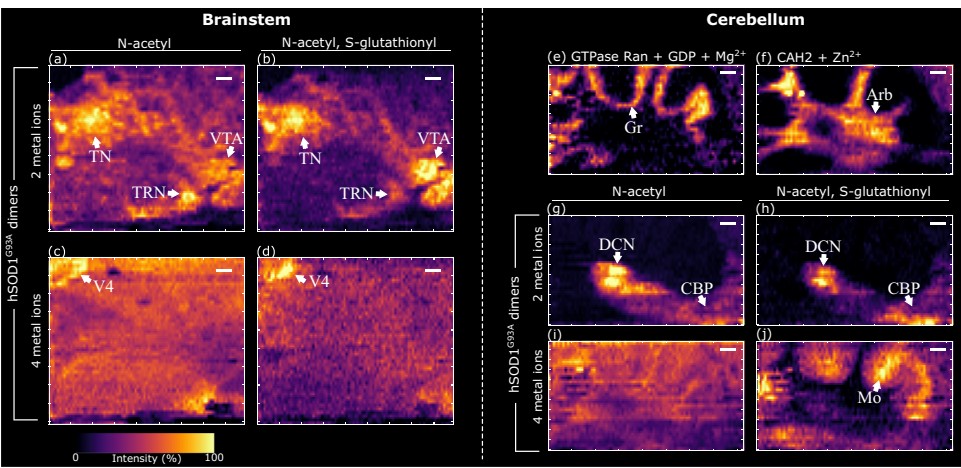

**Fig. 4 | High-resolution ion images illustrating protein complex distributions due to PTMs in the brainstem and cerebellum.** Brainstem ion images for hSOD1[G93A] dimers binding two metal ions and comprised of **a** two N-acetyl subunits and **b** one N-acetyl subunit and one N-acetyl, S-glutathionyl subunit; both dimers are distributed in cranial nuclei. hSOD1[G93A] dimers binding four metal ions and comprised of **c** two N-acetyl subunits and **d** one N-acetyl subunit and one N-acetyl, S-glutathionyl subunit. Cerebellum ion images for (**e**) GTPase Ran + GDP + Mg²⁺ complex in the granular layer and **f** CAH2 + Zn²⁺ complex in the abor vitae (Arb) are shown to assist with orientation of the hSOD1[G93A] ion images. hSOD1[G93A] dimers binding two metal ions and comprised of (**g**) two N-acetyl subunits and **h** one N-acetyl subunit and one N-acetyl, S-glutathionyl subunit are both localized to grey matter of the deep cerebellar nuclei (DCN) and cerebellar peduncles (CBP). hSOD1[G93A] dimers binding four metal ions and comprised of (**i**) two N-acetyl subunits and **j** one N-acetyl subunit and one N-acetyl, S-glutathionyl subunit. The former is ubiquitous and the latter abundant in the molecular layer. Ion images are composites of multiple charge states; hSOD1[G93A] dimers (11⁺, 10⁺, 9⁺), GTPase Ran (9⁺ and 8⁺), CAH2 (9⁺ and 8⁺). Presented with a linear intensity scale, 1 order of linear interpolation, and TIC normalized. TN tegmental nuclei, TRN tegmental reticular nucleus, VTA ventral tegmental area, V4 fourth ventricle, Gr granular layer, Arb arbor vitae, DCN deep cerebellar nuclei, CBP cerebellar peduncles, Mo molecular layer. Scale bars = 200 μm.

regions of neuronal loss in cases of Parkinson's disease which is linked with oxidative stress[47].

Dimers bound to 3 metal ions exhibit similar localization to the 2 metal-bound dimers, which suggests that the metal deficiency is partially due to inefficient copper incorporation rather than premature disulfide formation exclusively. Low abundance or efficacy of CCS within affected motor regions could result in an increase in the relative abundance of Cu-deficient hSOD1[G93A] dimers. It is possible that a proportion of the two metal-bound dimers are disulfide-reduced species, as suggested by Tajiri et al.[43]. In that in vitro study, zinc ions in excess could not be incorporated into the disulfide-reduced dimers beyond one zinc ion per monomer, whereas disulfide-intact dimers were able to bind up to five zinc ions. As the 3 metal-bound dimers exhibit ventral horn and brainstem nuclei localization, it is possible these are dimers with 3 zinc ions.

An alternative explanation could be low availability of Cu ions, eventually leading to the accumulation of Cu-deficient SOD1 species, both soluble and aggregated[7,48]. Studies in hSOD1[G93A] and hSOD1[G37R] mice where the Cu-containing therapeutic Cu(atsm) was administered increased SOD1 enzymatic activity by populating Cu-binding sites[26], resulting in prolonged survival and protection of motor neurons in the CNS[26,49,50]. Recent work in sporadic ALS cases in humans also implicates aberrant copper bioavailability in disease progression by broadly affecting cuproenzymes[51]. It is also possible that Zn-deficient hSOD1[G93A] species are present, previously reported to cause motor neuron apoptosis in mice[52,53] and reported in human spinal cord for various cases of ALS[54]. In summary, the affected regions of CNS tissue inevitably contain a mixed population of Zn and Cu-bound, but overall metal-deficient, hSOD1[G93A] complexes. Separation of complexes with different metals and disulfide bond states by mass alone is still challenging; further work will require high-resolution MS in tandem with gas phase fractionation methods, e.g., ion mobility spectrometry or ion–ion reactions[55,56].

## Discussion

Central nervous system tissue sections from transgenic mice expressing human SOD1 were imaged by native ambient MSI. The disease model, hSOD1[G93A], showed abundance of metal-deficient hSOD1[G93A] complexes in motor-associated structures of the CNS. The distribution indicates a relationship between the cellular environment of the affected motor regions and hSOD1[G93A] existing in a demetalated state. It may be that the demetalated hSOD1[G93A] is misfolded and hinders metal ion binding, or demetalation may be a precursor of further misfolding and aggregation. Monomerisation and glutathionylation were not found to influence spatial localization of metal-deficient complexes; metal-binding state remained the key factor. The cause of the loss or omission of metal ions from hSOD1[G93A] complexes in these tissue structures remains to be explained. We hypothesize that a property of the local cellular environment, e.g., metal ion concentration (insufficient Cu in CNS tissues of the hSOD1[G37R] model[46]), availability of CCS, pH, etc., is responsible for hSOD1[G93A] metal ion deficiency.

Current practical limitations prevent elucidation of which metal ions are bound and the status of the disulfide bond in hSOD1[G93A] complexes during a native ambient MS imaging experiment. MS approaches using ultra-high mass resolving powers are largely incompatible with MSI, although methods have been demonstrated which might be adaptable in the future[55,56]. Imaging methods using a mode of tandem mass spectrometry approaches or ion mobility spectrometry may be able to reveal further information. Nevertheless, untargeted visualization of protein distributions with specificity for their non-covalent and covalent modifications is a distinctive capability provided by NAMS for neurodegenerative disease pathology.

Overall, these results have implications for understanding the role of SOD1 toxic gain of function in ALS, which is particularly relevant in the context of therapeutics which reduce mutant SOD1 levels in ALS patient CNS, such as Tofersen[57,58]. The techniques developed here show that demetalation is a key pathological change observed for hSOD1[G93A] and may help to define whether this mechanism contributes to a broader population of sporadic ALS. Further work will focus on the application of native ambient mass spectrometry imaging of the various hSOD1 species over the time-course of the development of pathology in the model, expansion to additional models and human cases, and continued development of the imaging technology.

## Methods

### Ethics statement

All studies using animals were carried out in accordance with the UK Animals (Scientific Procedures) Act 1986, and all procedures were carried out under a Home Office project licence, reviewed, and approved by the local ethics committee (University of Sheffield Animal Welfare and Ethical Review Body). All animal maintenance and day-to-day care was carried out in line with Home Office Code of Practice for Housing and Care of Animals Used in Scientific Procedures.

### Materials

MS-grade water (catalog number 10095164) was purchased from Fisher Scientific (Loughborough, UK). HPLC-grade ammonium acetate (catalog number 15513351) was bought from J.T. Baker (Deventer, Netherlands). The detergent $C_8E_4$ (catalog number T3394), SOD1 and IL-2 primers (detailed below), agarose powder (catalog number A9539), and ethidium bromide solution (catalog number E1510) were bought from Sigma-Aldrich (Gillingham, UK). Mass spectrometer calibration was performed with FlexMix (catalog number A39239, Thermo Fisher, San Jose, CA). Nitrogen (>99.995%) and helium (>99.996%) gases used on the mass spectrometer were obtained from BOC (Guildford, UK). QuickExtract DNA extraction solution (catalog number QE09050) was purchased from Cambio Ltd (Cambridge, UK). Master mix (catalog number SBD-04-11-00115) was purchased from Thistle Scientific (Rugby, UK).

### Transgenic mice

SOD1G93A C57BL/6 transgenic mice, originally B6SJL-Tg (SOD1G93A) 1Gur/J (stock number 002726) mice were obtained from Jackson Laboratory and backcrossed onto the C57BL/6 OlaHsd background (Harlan UK, C57BL/6J OlaHsd) for at least 20 generations to create the SOD1G93A C57BL/6 transgenic line on an inbred genetic background[59]. The SOD1[G93A] transgene is maintained as a hemizygous trait by breeding hemizygous males with wild-type female mice (C57BL/6J OlaHsd, Harlan UK). Breeding was performed in-house at the University of Sheffield.

Human SOD1[wt] mice (B6.Cg-Tg(SOD1)2Gur/J) were obtained from the Jackson Laboratory, strain number 02298[20], and these were backcrossed onto the C57BL/6 OlaHsd background (Harlan UK, C57BL/6J OlaHsd) for at least 20 generations. The colony was maintained in-house by breeding female transgenic mice with C57LB/6J mice. This transgenic strain carries the normal allele of human SOD1 and does not carry the SOD1[G93A] transgene.

Mice were housed in groups of between 2 and 5, with one plastic house provided per cage, sawdust (Datesand) was used to cover the floor of the cage and paper wool bedding (Datesand) was provided as nesting material. All animals were housed in rooms maintained at a temperature of 21 °C and with a 12-h light/dark cycle. Food (standard rodent diet 2018, Evigo) and water were provided ad libitum, with weekly water changes.

Mice were earclipped for the purpose of identification and genotyping. Extraction of DNA from the earclip was performed using 20 µl of Quickextract (Lucigen), with incubation at 65 °C for 15 min followed by 98 °C for 2 min. Genotyping for hSOD1[G93A] and hSOD1[wt] mice was carried out using the same protocol. PCRs for genotyping were performed in a 10 µl volume of master mix containing 5x FIREPol Master Mix with 7.5 mM $MgCl_2$ (Thistle Scientific, UK, SBD-04-11-00115), 150 nmol each of human SOD1 primers (forward 5′-CATCAGCCCTAATCCATCTGA-3′, reverse 5′-CGCGACTAACAATCAAAGTGA-3′) and control interleukin-2 receptor (IL-2R) primers (forward 5′-CTAGGCCACAGAATTGAAA GATCT-3′, reverse 5′-GTAGGTGGAAATTCTAGCATCATC-3′), nuclease-free water, and 0.5 µl of DNA sample. Gel electrophoresis was performed on PCR products on a 2% agarose gel with 1 µl ethidium bromide solution per 100 ml, IL-2R products were visualized at 324 bp and human SOD1, if present in the case of transgenic animals, was present at 236 bp.

### Brain and spinal cord tissue

**Fresh-frozen tissue.** Brains and spinal cords from transgenic mice (120–130 days old, see Supplementary Table S1, Supplementary Information) expressing a mutant human isoform of SOD1, hSOD1[G93A], were snap-frozen over dry ice on foil and then stored at −80 °C. Control brains and spinal cords from transgenic mice (120–180 days old, Supplementary Table S1, Supplementary Information) expressing the human wild-type protein (hSOD[wt]) were prepared in the same way. Lumbar spinal cord tissue was processed as follows; the lumbar enlargement was dissected from the cord and bisected. Lumbar cord coronal sections were collected at 10 µm thickness outwards from the bisection point. Serial sagittal brain sections of 10 µm thickness were prepared from the left-brain hemisphere, cutting outwards from the midline. Cryosections were prepared with a CM1810 Cryostat (Leica Microsystems, Wetzlar, Germany) and thaw mounted to glass microscope slides before storage at −80 °C until analysis. The tissue was not washed or fixed before native ambient mass spectrometry imaging.

**Fixed tissue.** formalin-fixed paraffin-embedded (FFPE) lumbar spinal cord was prepared from hSOD1[G93A] transgenic ($N = 5$) and mSOD1[wt] ($N = 5$) mice at $120 \pm 3$ days of age. Lumbar spinal cord (L3/4) was perfused under terminal anesthesia with PBS followed by 4% paraformaldehyde in PBS, then were processed and embedded in wax blocks, sectioned at 10 µm thickness on a microtome and every fifth section arrayed on glass slides. A total of 54 sections were analyzed for motor neurons (hSOD1[G93A] $N = 28$; mSOD1[wt] $N = 36$). FFPE-embedded brainstems were similarly obtained from mice of the same genotypes (hSOD1[G93A] $N = 3$; mSOD1[wt] $N = 2$) for a total of 27 sections (hSOD1[G93A] $N = 15$, mSOD1[wt] $N = 12$).

### Native ambient mass spectrometry imaging (MSI)

Native ambient MSI was performed using a home-built nano-DESI ion source attached to an Orbitrap Eclipse mass spectrometer (Thermo Fisher Scientific; Fig. 1) equipped with the high mass range (HMR[n]), electron transfer dissociation (ETD) and proton transfer charge reduction (PTCR) options[21,24]. The analytical solvent system was aqueous ammonium acetate (200 mM) with 0.125% of the detergent $C_8E_4$ added by volume. Solvent flow rate was optimized to between 0.5 and 2.0 µL/min, and spray voltage was between 800 and 1300 V. Electrospray stability was assessed using the linear ion trap (LIT) mass analyser by monitoring $m/z$ 307.21 until an RSD% <15% was achieved by tuning flow rate, electrospray voltage and emitter position.

The mass spectrometer was operated in intact protein mode. Typical source backing pressure was 2.3 Torr. The source ion transfer tube temperature was 275–300 °C. The RF lens was set to 120%. Source dissociation voltage (SDV) was set to 80 V, and the source compensation value (SCV) was tuned for each experiment to between 2.5 and 3.2%, as day-to-day differences in ambient temperature and chamber pressures in the mass spectrometer affect the optimal value. The importance of these settings has been discussed elsewhere[23,60,61]. The ion routing multipole (IRM) was set to a pressure of 20 mTorr using $N_{2(g)}$. All ion images were collected using a selected ion monitoring (SIM) method (depicted in Fig. 1). Ions were accumulated in the IRM followed by transfer to the LIT. An isolation window of $m/z$ 3197 ± 625 (brain) or $m/z$ 3000 ± 1000 (spinal cord) was applied before transmission of ions to the orbitrap analyzer for mass-to-charge ratio ($m/z$) measurement. Supplementary collisional activation and an averaging of 5 microscans was used to improve S/N for spinal cord images. The LIT damping gas (He$_{(g)}$) pressure was $3.5 \times 10^{-5}$ Torr. The automatic gain control target was set to 10,000% ($5 \times 10^6$ charges) with a maximum injection time of 750 ms. The orbitrap analyzer was operated at a resolution setting of 7500 FWHM (at $m/z$ 200, 16 ms transient length). Proceeding with imaging experiments was conditional on achieving a signal intensity $>1 \times 10^4$ (normalized level) for holo-hSOD1[G93A] dimer (10+ charge state) under these instrument conditions.

For full brain section (hSOD1$^{wt}$ $N = 3$, hSOD1$^{G93A}$ $N = 3$) imaging, the tissue was moved under the nano-DESI probe at 20 μm/s with a line step of 200 μm. For spatially targeted high-resolution brain MSI (hSOD1$^{wt}$ $N = 1$, hSOD1$^{G93A}$ $N = 1$), the movement velocity was 3 μm/s with a line step of 50 μm. Full spinal cord sections (hSOD1$^{wt}$ $N = 2$, hSOD1$^{G93A}$ $N = 2$) were imaged at 8 μm/s with a line step of 100 μm. High-resolution spinal cord MSI (hSOD1$^{G93A}$ $N = 1$) was performed at 3 μm/s with a line step of 30 μm.

Mass spectrometer cycle time was ~1 s with automated analysis of each brain section requiring ~6 h. High-resolution images were analyzed from separate serial sections and required ~11 h to record. Spinal cord analysis required ~3 h per section, and 3 h for high-resolution analysis of the 1 mm × 0.69 mm region.

### Ion image processing

Ion image files were generated by conversion of the Thermo raw files acquired for each line scan to a single imzML file using Firefly (v.3.2.0.23, Prosolia, Inc., Indianapolis, IN). Pixels in full brain section ion images were composed of ten seconds of summed MS scans representing 200 × 200 μm (0.04 mm$^2$) of tissue and are referred to as "moderate resolution MSI". Similarly, spinal cord image pixels were composed of scans representing 100 × 100 μm of tissue (~0.01 mm$^2$). Pixels in spatially targeted, high-resolution MSI had pixel dimensions of 30 × 50 μm (in brain, representing ~0.0015 mm$^2$ of tissue) and 14 × 30 μm (in spinal cord, ~0.00042 mm$^2$). Ion images were processed in MSiReader (v 1.02, North Carolina State University)[62], had 1× linear interpolation applied, and a linear intensity scale normalized to the most intense pixel in the image at the selected $m/z$ ± 0.1. Ion images for each charge state of the hSOD1 dimers, [Arf1+GDP], [Arf3+GDP] and [CAH2 + Zn$^{2+}$] complexes, and β-synuclein were summed using a MATLAB script (available from https://github.com/coopergroup-massspec/sum_matlab_figures, and since facilitated by a high throughput imaging module in UniDec version 6.0.4[63]) to create images composed of all detected signal for each complex. (Ion images for monomeric SOD1 comprise a single charge state (6 +)). Constituent ion images for images comprising multiple protein charge states are found in the Supplementary Information and referenced specifically in the main text. Non-SOD1 protein images were inspected to validate analytical performance between experiments (Supplementary Fig. S30, Supplementary Information).

### Top-down MS analysis

For native top-down MS of protein–metal complexes, tissue was sampled by nano-DESI using aqueous ammonium acetate (200 mM) with 0.125% of the detergent C$_8$E$_4$ added by volume. Protein–metal complex ions were subject to gas phase dissociation using beam-type collisional activation (HCD) in N$_{2(g)}$ or resonant collision-induced dissociation (CID) in He$_{(g)}$. The IRM pressure was 20 mTorr and the orbitrap resolution was 240,000 (FWHM at $m/z$ 200, 512 ms transient). Normalized collision energy (NCE) was typically set to 30–35% for dissociation of non-covalent SOD1 complexes and 39–45% for top-down protein sequencing (10+ charge state). Top-down analysis using electron transfer dissociation with supplemental collision activation (EThcD) was performed with reaction time 10–18 ms, and supplemental collision energy of 10–12%.

Top-down analysis of denatured proteins was performed using contact LESA of tissue with a Triversa Nanomate robot (Advion Biosciences, Ithaca, NY)[64]. The solvent system was acetonitrile/water +0.1% formic acid (1:1 v/v) to dissociate metal ions and unfold hSOD1 in solution. Protein ions were $m/z$ selected using the quadrupole mass filter then fragmented by collisional activation (HCD; NCE 38%) in the IRM (pressure = 8 mTorr). Orbitrap resolution was 240,000 (FWHM at $m/z$ 200).

Fragment ion spectra were automatically matched to theoretical fragment ions with a tolerance of 20 ppm for the hSOD1$^{G93A}$ sequence using Prosight PC (version 4.1, Thermo), followed by manual interpretation and validation.

### Spectral deconvolution

Mass spectra were deconvoluted with UniDec (version 6.0.4)[65] to determine protein complex mass from multiple protein ion charge states from non-isotopically resolved mass spectra. Parameters other than those described in Supplementary Table S7, Supplementary Information were left at default values.

### Histology

**Nissl staining procedure.** Tissue sections were allowed to warm to room temperature before staining. Slides were placed into 95% alcohol for 5 min, followed by 70% alcohol for a further 5 min, and finally washed in tap water. 0.1% Cresyl fast violet solution was first filtered, and slides were then stained for 20 min. Stained sections were differentiated in acetic acid for around 4 s and then washed in 95% alcohol. This differentiation time allowed for sufficient separation between the background and Nissl-stained cells of interest. At this point the sections could be viewed under a microscope to verify the background separation, and the slides could be returned to acetic acid for further differentiation if necessary. Finally, slides were dehydrated in absolute alcohol for 2 × 5 min before being cleared in xylene and coverslips mounted with DPX.

### Optical scanning and tissue annotation

Stained tissue sections were scanned at ×20–40 magnification using a Hamamatsu Nanozoomer XR. Scanned sections were visualized with QuPath version 0.4.1[66]. Tissue was annotated with reference to the Allen Adult Mouse Atlas[67].

### Statistical analysis

**Evaluation of relative abundances of SOD1 complexes (Figs. 2o and 3o).** The mean signal intensity of each complex (in dorsal horns; $n = 4$, ventral horns; $n = 4$, XII; $n = 3$, frontal cortex; $n = 3$) for each genotype was averaged. Charge states 11 +, 10+ and 9+ were included for SOD1 dimers; for monomers, 6+ and 5 +. The 10+ dimers and 5+ monomers overlap in $m/z$ in the imaging datasets, but high-resolution mass spectrometry was used to estimate a 52% contribution to signal intensity by the 10+ dimers (Supplementary Fig. S31, Supplementary Information). The statistical significance of signal intensity differences between VH and DH, and XII and frontal cortex, were evaluated by Student's unpaired $T$ test.

**Motor neuron counts in the ventral horn (Fig. 2p).** Nissl-stained and lumbar ventral horn motor neurons were counted from intact hemi-sections from each genotype (hSOD1$^{G93A}$ sections = 28, mSOD$^{wt}$ sections = 36). Motor neurons were distinguished based on location (ventral horn) and morphological criteria as follows: large multipolar cells, cell body with a size of at least 25 μm in any one dimension, a distinct nucleus and nucleolus. An average number of motor neurons were then determined for each mouse ($N = 5$ hSOD1$^{G93A}$ transgenic mice and $N = 5$ non-transgenic mice) genotype and analyzed by unpaired $T$ test.

**Motor neuron counts in the hypoglossal nucleus (XII, (Fig. 3p).** Motor neurons were counted, as for the spinal cord, in FFPE brainstem sections (hSOD1$^{G93A}$ sections = 15, mSOD$^{wt}$ sections = 12) for each mouse genotype (hSOD1$^{G93A}$ $N = 3$, mSOD$^{wt}$ $N = 2$).

### Reporting summary

Further information on research design is available in the Nature Portfolio Reporting Summary linked to this article.

## Data availability

The mass spectrometry and optical imaging data generated in this study have been deposited in the University of Birmingham

Institutional Research Archive under accession code [https://doi.org/10.25500/edata.bham.00001123]. Processed data are also available in the same archive. Mass spectrometry data have also been deposited to the ProteomeXchange Consortium via the PRIDE partner repository with the dataset identifier PXD053247 [https://www.ebi.ac.uk/pride/archive/projects/PXD053247]. The signal intensity and motor neuron count data generated in this study are provided in the Source Data file. Source data are provided with this paper.

## Code availability

Code for the generation of composite ion images[23] is available from https://github.com/coopergroup-massspec/sum_matlab_figures and https://doi.org/10.5281/zenodo.12572632.

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

## Acknowledgements

O.J.H. and H.J.C. acknowledge EPSRC (EP/S002979/1) for funding. The Orbitrap Eclipse mass spectrometer used in this work was funded by BBSRC (BB/S019456/1).

## Author contributions

O.J.H.: study design, tissue preparation, mass spectrometry, data processing, manuscript draft, and review. T.R.W.: tissue collection and histology and manuscript review. R.J.M.: study design, tissue collection, manuscript review. H.J.C.: study design, data analysis, manuscript draft, and review.

## Competing interests

The authors declare no competing interests.
