## [Peer Review File · Nature Communications]

Reviewers' Comments:

Reviewer #1:

Remarks to the Author:

This is a major technological advance that provides important new insights into how SOD mutants cause ALS. This provides much needed data on the endogenous differences in metal content, dimerization and glutathionylation in motor neuron regions of cortex and cerebellum.

The results showing low concentration of metal-deficient dimers in wtSOD (which do not cause ALS in mice normally) and G93A SOD mice in brain regions known to be affected by the ALS disease-causing processes. It would be useful to make a summary bar graph from Figure S16b to show in the main text of the paper, because this shows the importance of native MS imaging to differentiating wt from mutant SOD in disease sensitive brain regions.

The ability of these methods to measure two glutathionylation events on SOD dimers with metal cofactor binding in Figure S17 is impressive and might be highlighted more in the main text. Also the figure legend mentions zinc containing carbonic anhydrase, but not shown. Carbonic anhydrase is closer to 29kDa and the left of the x axis ends at 31 kDa.

There are studies strongly supporting the stabilization of SOD1 accelerates the progression of motor neuron death in culture as well as in transgenic mice (see Garner et al. Cu,Zn-superoxide dismutase increases toxicity of mutant and zinc-deficient superoxide dismutase by enhancing protein stability. *J Biol Chem.* 2010 Oct 29;285(44):33885-97. doi: 10.1074/jbc.M110.118901). Notably, mutating C111 to serine greatly increased the toxicity of copper-containing, zinc-deficient SOD. Cu,ZnSOD mixed with Cu(-)SOD was also much more toxic. ¹¹SEP A major result is the measurement of monomers and dimers of SOD present in the gas phase after ionization. However, this will not reflect the presence of SOD in tissues. Each zinc-binding loop of SOD forms ~37% of the dimer interface and the zinc binding loop also contains cysteine-57, which forms one half of the disulfide loop (See Roberts et al. *J. Mol. Biol.* 2007 373: 877-90). Thus, metal binding and disulfide bond are intimately related to the stability of the dimer interface. Mixed states of partially folded and metal-content will result in weak bonds that will appear as monomers during imaging. It will be important to point out that measured monomers likely reflect populations of SOD proteoforms with weakened dimer interfaces.

It would helpful to make a figure/cartoon in supplemental information to illustrate the multiplicity of SOD proteoforms that could give rise to the observed monomeric and dimeric states observed by the mass spec. This will help readers who work more with mice than SOD and MS to understand the underlying complexity.

Page 5. "i.e., misfolded SOD1 results in formation of monomers which subsequently aggregate into insoluble inclusion bodies." With protein aggregation is a widely cited cause of neurodegeneration given in textbooks, there is a large body of work that argues SOD aggregation does not necessarily correspond to the disease process. The ALS field has converged on a misfolded form of SOD1 being the cause of disease, without the need for aggregation (e.g. the Garner reference given above). Monomerization is experimentally known to promote metal loss, but the metal-deficient monomers are still in a rapid equilibrium with other metal-containing SODs. This may explain why crossing G93A SOD mice with wild type SOD mice results in much faster disease progression. While beyond the scope of this already-packed paper, the F1 crossing of the two SOD transgenics would be a nice extension of this work, worthy of another publication.

Minor points

It may be helpful to spell out MSI and MS imaging, or better native MS imaging. It would also help differentiate from Maldi Imaging, which is widely practiced and recognized. This will highlight the major differentiator for the paper.

Was it possible to use ETD on the Eclipse Orbitrap MS for electron-based fragmentation. With electrospray, it can cleave just before C57 to yield fragments containing copper or zinc. Sensitivity will certainly be a limitation here.

Most of the pathology of ALS happens in the ventral spinal cord and even in the ventral roots. It would help to explain why the present study focused on brain. Having dissected mouse spinal cords, the difficulties of working with such small and delicate tissues is appreciated, as well as the need for many more mice. An explanation of this and the need for future work should be stated more explicitly.

Reviewer #2:

Remarks to the Author:

Despite the passing of several decades since the discovery that mutant SOD1 causes ALS, further insight to how mutant SOD1 contributes to neuronal death is still needed. In their manuscript, Hale and colleagues provide new insight from the perspective of spatially resolved mass spectrometry detection of SOD1 in mouse brain samples. The key strength of this work is that the mass spec imaging methodology they utilised informs on metallation state of the protein. Metallation state of SOD1 is recognised as an important determinant of the protein's role in neurodegeneration. Therapeutic agents that modulate the metallation state of SOD1 have been developed. Showing that metal-deficient/demetallated SOD1 spatially associates with neuronal loss in the central nervous system would have a strong impact in the field.

Additional strengths of this study are the use of mice that over-express WT human SOD1 as a control reference point, and assessment of glutathionylated SOD1 as a relevant post-translational modification.

There are two key aspects to this study. First, the achievements made with respect to application of the mass spectrometry imaging technology. Second, interpretation of the results generated with respect to ALS pathology. With a focus on the latter, I have several comments/suggestions:

1. Emphasising the results for metal-deficient/demetallated SOD1 as being associated with ALS pathology (e.g., title and abstract) requires clearer evidence for relevant pathology (neuron loss) in the regions where evidence for metal-deficient/demetallated SOD1 is strongest. To this end, presentation of more of the quantified data is necessary. The authors need to show that the regions in which metal-deficient SOD1 is most evident are also the regions in which neuronal loss is evident. The point of comparison in each instance needs to be the corresponding brain region in the hSOD1WT mice. As it stands, the only quantified evidence for an accumulation of metal-deficient/demetallated SOD1 is presented in Fig. S16 where it is shown there is more metal-deficient/demetallated SOD1 in the hSOD1G93A XII when compared to the hSOD1G93A full brain. To support an argument for an association between metal-deficient/demetallated SOD1 and relevant pathology, it needs to be shown that there is more metal-deficient/demetallated SOD1 in the hSOD1G93A XII when compared to the hWTSOD1 XII and this needs to be associated with fewer neurons in the hSOD1G93A XII when compared to the hWTSOD1 XII. This data set for quantified metal-deficient/demetallated SOD1 being associated with quantified neuronal loss should be included in the main figures. (Negative data for the involvement of glutathionylation could be relegated to the supplementary information.)

2. Related to point #1, rather than propose spinal cord analysis for future expansion of the work, the inclusion of spinal cord data for metal-deficient/demetallated SOD1 in the ventral horn of the spinal cord, along with motor neuron numbers, is essential for this manuscript. This is the region of the central nervous system where neuronal loss is most pronounced in this model.

3. Clearer definition of the term "heterodimer" will be beneficial on page 8. Others have used the term previously to describe a dimer of SOD1 comprised of one WT monomer and one mutant monomer. Based on the primary focus of this study (metal binding), an alternate use could involve a dimer that contains one holo-monomer and one apo- or partially metallated monomer. Neither of

these matches the use that is intended in this part of the manuscript which is glutathionylation state.

4. The caveat "...presumably by populating Cu-binding sites" needs to be revised on page 11. One of the studies cited (#26) confirmed via tracking isotopically labelled Cu that Cu(atSm) did indeed deliver Cu to SOD1.

5. A hypothesis is presented at the end of the first paragraph in the Conclusions section. With respect to the metal ion concentration part of this hypothesis, the authors need to consider and cite a relevant paper (PMID: 27357743). Although this paper addresses the concept of copper availability at the tissue/organ level, it may be possible that within the central nervous system a similar availability scenario could play out at a more region-specific level. The authors' mass spectrometry imaging data could support such speculation.

6. A role for insufficiently metalated SOD1 being associated with neuronal loss is described in a study that assessed human brain from cases of Parkinson's disease (PMID: 28527045). This association was confirmed to not involve mutant SOD1 (PMID: 28527045). These studies need to be cited. They provide supportive insight to whether neuronal loss associated with metal-deficient/demetalated SOD1 can occur independently of SOD1 mutations.

7. An additional paper from the same team as above examined human spinal cord from cases of ALS (PMID: 35512359). This paper should also be cited due to evidence provided for the presence of SOD1 in the human tissue with a compromised metalation state.

8. The methods section needs to describe how motor neurons were identified relative to other Nissl-positive structures.

Reviewer #3:

Remarks to the Author:

The manuscript by Hale et al. employs native ambient mass spectrometry imaging technique to examine spatial distributions of SOD1 protein-metal complexes in SOD1 mutant mice, with the goal to make some correlation between metalation states, protein oligomerization and development of pathology. While a very interesting study, the report suffers from several issues. The method of nanoDESI-MS imaging for protein MS imaging is not new, several previously published papers have demonstrated the capability, with a few listed below:

J. Am. Chem. Soc. 2022, 144, 5, 2120–2128 (from the authors' group);

Angew. Chem., Int. Ed. 2022, 61, e202200721 (Julia Laskin group).

Therefore, the major evaluation criteria would be the novel insight of biology. However, there are some major issues about the current data that make their findings in this manuscript questionable and need to be carefully addressed:

1. Proper control experiments are needed to claim the native metal binding is not affected by the existence of high concentrations of other prevalent metal cations (Na⁺, K⁺, NH₄⁺, etc.) Could test native metal-binding protein by spiking different salts. Although the authors indicated that m/z values of the detected MS1 peaks (e.g., Fig S7) matched with different binding numbers of Cu/Zn to the dimers. It is also very likely some or even a large portion of these comes from 3n*Na⁺ adduction as Na⁺ is prevalent in biological tissues. E.g., the mass increment of [M+3*Na+n*H] to [M+(n+3)*H] is 66 Da, very close to the addition of Cu or Zn, of which the isotopic mass is 63~68 Da. And this is very likely to be true if looking at all listed HCD MS2 spectra (Fig. S8-S10). From zoom-in spectra of the monomeric fragment ions, 2 small peaks almost equally distributed (~22 Da mass difference) between 2 adjacent monomers indicated different metalated states could be clearly seen, which may correspond to +1*Na⁺ adduction and +2*Na⁺ adduction. If the metalation is only Cu/Zn, the monomeric products dissociated from dimers should only have mass differences over 60 Da and those peaks with 22 Da mass differences should not be observed. Please comment on the formation of these small peaks in HCD MS2 spectra.

2. The authors indicated that "Ion images for monomeric hSOD1wt and hSOD1G93A showed similar spatial distributions to the dimers of equivalent metal-binding state" and concluded that "demetalation is a key pathological change observed for SOD1". These still need more critical

examination. It is important to diminish overlapping intensities of dimers at 2x charge state as monomers. It is good that the authors noticed it and attempted to omit 10+ dimers and 5+ monomers and only used 11+ and 9+ for dimers and 6+ for monomers. However, from Fig. S7, 10+ is actually the most abundant charge state for dimers. 6+ may not be determined as the most abundant charge state for monomers and 5+ could be very likely the most. This elimination would largely obscure the quantitative accuracy when dimer/monomer ratios are investigated as they are comparing things not at the same criteria, integrating only 1 charge state for monomer and 2 charge states for dimers. Besides, 6+ monomer still overlaps with 12+ dimer. How to make ratio calculations as monomer intensity cannot be obtained reliably? This also places concerns about the MS images of monomers. The co-localization of monomer (sum peak intensity at 6+ charge state only) could be false-positive and be attributed to the presence of dimer at the same m/z (at 12+ charge state).

3. Furthermore, there are also concerns about the proportion of in-source dissociation of dimers to monomers and if that would be influential to the MS imaging of monomers. From the manuscript, such evaluation has not been done in their nanoDESI-MS setup. This could be done with protein standards. Can the authors provide more results and comments on this aspect?

4. This work does not have sufficient evidence to support the conclusion that only demetalation, not monomerization, correlates with SOD1-G93A transgenic mouse pathology.

5. Based on the mass spectra shown in Figure S7, the signal intensities of hSOD1(s) were generally much higher in the G93A mouse brain, and the signal-to-noise was very poor in the WT mouse. It looks like the results shown in Figure 2 were based on Figure S7. The authors need to better demonstrate the reproducibility of the method, and address the variation issue. Some additional data about the reproducibility and stability of the method would help to address these concerns.

6. I would suggest the authors to add some protein standards in the DESI solvent to nominalize the data detected from the WT and G93A mice brain.

7. Figure S14 d-f, higher signals of hSOD1 were detected from the area off the tissue. This observation showed that there is severe chemical delocalization for the tissue being imaged, further casting doubt of the conclusions drawn from this data.

8. On page 10, since the Zn-binding might inhibit Cu-ion incorporation and future maturation, the authors are expected to highlight the Zinc and Copper binding separately in the deconvolution progress. Specifically, do holo-monomers consist of the same ratio of Zn and Cu as the holo-dimers? If there is an unstated difference in preference monomerization towards Zn²⁺, then we cannot reach this conclusion by merely comparing the intensity of the monomer peak. This is highly confusing as Supplementary figures 7-11 & 17 are all annotated as "Zn/Cu ions".

In comparison, there is another article published in 2022

(<https://doi.org/10.1093/brain/awac165>), on page 3119, the author specifically clarified that

"Zinc-deficient mature SOD1 is unlikely to undergo self-assembly from a native-like dimeric state", and in this in-vivo testing, the Cu:Zn ratio within SOD1 aggregate is carefully measured.

9. For the PTM analysis, glutathionylation at Cys111 shown in article is previously found strongly associated with structural stability. But there are also many other phosphorylation, acetylation, and ubiquitination sites(<https://doi.org/10.1016/j.redox.2019.101270>). The authors are expected to examine how much extent the other PTMs may have an effect on unfolding.

Response to Reviewers' comments:

Reviewer #1: It would be useful to make a summary bar graph from Figure S16b to show in the main text of the paper, because this shows the importance of native MS imaging to differentiating wt from mutant SOD in disease sensitive brain regions.

Response: As described above, we have conducted additional experiments on spinal cord tissue as requested by Reviewer 2 as this is the region of the central nervous system most affected in this model. The revised manuscript includes a chart (Fig 2o) comparing data from the ventral (disease-sensitive) and dorsal horns of the spinal cord which similarly demonstrates the importance of native MS imaging. We performed a similar analysis for the brainstem, found in (new) Figure 3.

Reviewer #1: The ability of these methods to measure two glutathionylation events on SOD dimers with metal cofactor binding in Figure S17 is impressive and might be highlighted more in the main text. Also the figure legend mentions zinc containing carbonic anhydrase, but not shown. Carbonic anhydrase is closer to 29kDa and the left of the x axis ends at 31 kDa.

Response: To maintain a concise narrative following the additional experiments described above, details of the identification and characterisation of the various SOD1 complexes have been moved to the Supporting Information. A section focusing on glutathionylation remains in the main manuscript ("Evaluation of metal binding and PTMs"). In addition, original Figure S17 (now Figure S3) has been updated to show spectra from brainstem tissue with an expanded mass axis in (b). Labels in (a) and (b) have been clarified.

Reviewer #1: There are studies strongly supporting the stabilization of SOD1 accelerates the progression of motor neuron death in culture as well as in transgenic mice (see Garner et al. Cu,Zn-superoxide dismutase increases toxicity of mutant and zinc-deficient superoxide dismutase by enhancing protein stability. J Biol Chem. 2010 Oct 29;285(44):33885-97. doi: 10.1074/jbc.M110.118901). Notably, mutating C111 to serine greatly increased the toxicity of copper-containing, zinc-deficient SOD. Cu,ZnSOD mixed with Cu(-)SOD was also much more toxic. ^[LSEP] A major result is the measurement of monomers and dimers of SOD present in the gas phase after ionization. However, this will not reflect the presence of SOD in tissues. Each zinc-binding loop of SOD forms ~37% of the dimer interface and the zinc binding loop also contains cysteine-57, which forms one half of the disulfide loop (See Roberts et al. J. Mol. Biol. 2007 373: 877-90). Thus, metal binding and disulfide bond are intimately related to the stability of the dimer interface. Mixed states of partially folded and metal-content will result in weak bonds that will appear as monomers during imaging. It will be important to point out that measured monomers likely reflect populations of SOD proteoforms with weakened dimer interfaces.

Response: The reviewer suggests that dimers that are weakly-bound in the tissue will dissociate in the gas-phase and appear in the mass spectra as monomers. Our results suggest that in-source dissociation of dimers does not occur during the imaging method. We have included a new figure (Figure S11) which compares spectra obtained with the imaging (SIM) method and with collisional activation (HCD). Collisional dissociation of the metal-bound dimers is associated with characteristic signals corresponding to reorganisation of metal ions to non-biologically relevant complexes and neutral losses. These were not observed in the imaging (SIM) spectra. We conclude that there is no

significant dissociation of dimers → monomers during imaging and that the monomer/dimer signals are representative of the in-tissue equilibrium. The text below has been added to the new section in the Supporting Information “Identification and characterization of protein complexes”.

“To confirm that monomers detected directly from the tissue were endogenous and not the product of collisional activation within the mass spectrometer, the SIM-mode mass spectra were compared with HCD MS² spectra of the dimers (Figure S11, Supporting Information). Collisional activation is accompanied by characteristic metal ion rearrangement which was not observed in the SIM-mode mass spectra confirming that the monomers were present in the tissue. (Note, it is possible that in-solution dissociation of dimers may occur after extraction and prior to ionization; however, the timescale for this is < 1 s).”

Reviewer #1: It would helpful to make a figure/cartoon in supplemental information to illustrate the multiplicity of SOD proteoforms that could give rise to the observed monomeric and dimeric states observed by the mass spec. This will help readers who work more with mice than SOD and MS to understand the underlying complexity.

Response: Thank you for the suggestion. We have now included a supplementary figure (Figure S2) depicting the variety of protein ions in the experiments in this study, which we hope will help with understanding for readers less familiar with native mass spectrometry. In addition, we have added a supplementary table (Table S2) which summarises the molecular weights of the protein species observed in the imaging experiments.

Reviewer #1: Page 5. “i.e., misfolded SOD1 results in formation of monomers which subsequently aggregate into insoluble inclusion bodies.” With protein aggregation is a widely cited cause of neurodegeneration given in textbooks, there is a large body of work that argues SOD aggregation does not necessarily correspond to the disease process. The ALS field has converged on a misfolded form of SOD1 being the cause of disease, without the need for aggregation (e.g. the Garner reference given above). Monomerization is experimentally known to promote metal loss, but the metal-deficient monomers are still in a rapid equilibrium with other metal-containing SODs. This may explain why crossing G93A SOD mice with wild type SOD mice results in much faster disease progression. While beyond the scope of this already-packed paper, the F1 crossing of the two SOD transgenics would be a nice extension of this work, worthy of another publication.

Response: We agree that such an experiment would be an interesting extension of this work and well-suited to the capabilities of NAMS. Thank you for the suggestion. This is under consideration for a further step in this research,

Reviewer #1: It may be helpful to spell out MSI and MS imaging, or better native MS imaging. It would also help differentiate from Maldi Imaging, which is widely practiced and recognized. This will highlight the major differentiator for the paper.

Response: Thank you for the helpful comment. We have revised the usage to native ambient mass spectrometry imaging (MSI or MS imaging) throughout the manuscript.

Reviewer #1: Was it possible to use ETD on the Eclipse Orbitrap MS for electron-based fragmentation. With electrospray, it can cleave just before C57 to yield fragments containing copper or zinc. Sensitivity will certainly be a limitation here.

Response: *ET(hc)D MS/MS experiments were performed as suggested by the reviewer; however these did not yield information on metal ion binding. The only c-ions observed were close to the N-terminus, i.e., originated from cleavages at residues <36. No product ions > m/z 2000 were detected. These data have been included in the revised manuscript (Figure S9, Table S4).*

Reviewer #1: Most of the pathology of ALS happens in the ventral spinal cord and even in the ventral roots. It would help to explain why the present study focused on brain. Having dissected mouse spinal cords, the difficulties of working with such small and delicate tissues is appreciated, as well as the need for many more mice. An explanation of this and the need for future work should be stated more explicitly.

Response: *As described above, we have performed additional experiments to include analysis of the spinal cord. These experiments show that the metal-deficient hSOD1G93A exhibits specific localisation to the ventral horn, correlating with pathology.*

--

Reviewer #2: Emphasising the results for metal-deficient/demetalated SOD1 as being associated with ALS pathology (e.g., title and abstract) requires clearer evidence for relevant pathology (neuron loss) in the regions where evidence for metal-deficient/demetalated SOD1 is strongest. To this end, presentation of more of the quantified data is necessary. The authors need to show that the regions in which metal-deficient SOD1 is most evident are also the regions in which neuronal loss is evident. The point of comparison in each instance needs to be the corresponding brain region in the hSOD1WT mice. As it stands, the only quantified evidence for an accumulation of metal-deficient/demetalated SOD1 is presented in Fig. S16 where it is shown there is more metal-deficient/demetalated SOD1 in the hSOD1G93A XII when compared to the hSOD1G93A full brain. To support an argument for an association between metal-deficient/demetalated SOD1 and relevant pathology, it needs to be shown that there is more metal-deficient/demetalated SOD1 in the hSOD1G93A XII when compared to the hWTSOD1 XII and this needs to be associated with fewer neurons in the hSOD1G93A XII when compared to the hWTSOD1 XII. This data set for quantified metal-deficient/demetalated SOD1 being associated with quantified neuronal loss should be included in the main figures. (Negative data for the involvement of glutathionylation could be relegated to the supplementary information.)

Response: *We have performed additional experiments as suggested by the reviewer. Neuron counting has been performed in both brainstem (XII) and spinal cord (ventral horn and dorsal horn) samples. Owing to the integrity of the tissue, it was not possible to count the motor neurons in the fresh frozen G93A tissue. We have therefore included a comparison of FFPE G93A mouse spinal cord versus wild-type at 120 days of age to show the neuron loss. Analysis of mean MS signal intensities in cortex and XII regions (brain), and ventral and dorsal horns (spinal cord), correlate with neuronal loss. These data (MS and neuron counts) are included in Figures 2 and 3. Representative Nissl-stained sections of spinal cord are shown in Figure S18, Supporting Information.*

Reviewer #2: Related to point #1, rather than propose spinal cord analysis for future expansion of the work, the inclusion of spinal cord data for metal-deficient/demetalated SOD1 in the ventral horn of the spinal cord, along with motor neuron numbers, is essential for this manuscript. This is the region of the central nervous system where neuronal loss is most pronounced in this model.

Response: *As described above, analysis of the spinal cord is now included.*

Reviewer #2: Clearer definition of the term "heterodimer" will be beneficial on page 8. Others have used the term previously to describe a dimer of SOD1 comprised of one WT monomer and one mutant monomer. Based on the primary focus of this study (metal binding), an alternate use could involve a dimer that contains one holo-monomer and one apo- or partially metalated monomer. Neither of these matches the use that is intended in this part of the manuscript which is glutathionylation state.

Response: *Thank you for this comment. We have revised the terminology to avoid confusion. Dimers containing one S-glutathionylated hSOD1 subunit are now specifically referred to as "1GS-dimers".*

Reviewer #2: The caveat "...presumably by populating Cu-binding sites" needs to be revised on page 11. One of the studies cited (#26) confirmed via tracking isotopically labelled Cu that Cu(at5m) did indeed deliver Cu to SOD1.

Response: *Thank you for this comment. We have revised the sentence and attached the reference.*

Reviewer #2: A hypothesis is presented at the end of the first paragraph in the Conclusions section. With respect to the metal ion concentration part of this hypothesis, the authors need to consider and cite a relevant paper (PMID: 27357743). Although this paper addresses the concept of copper availability at the tissue/organ level, it may be possible that within the central nervous system a similar availability scenario could play out at a more region-specific level. The authors' mass spectrometry imaging data could support such speculation.

Response: *Thank you for the suggestion. We have added a comment and citation to the conclusions.*

Reviewer #2: A role for insufficiently metalated SOD1 being associated with neuronal loss is described in a study that assessed human brain from cases of Parkinson's disease (PMID: 28527045). This association was confirmed to not involve mutant SOD1 (PMID: 28527045). These studies need to be cited. They provide supportive insight to whether neuronal loss associated with metal-deficient/demetalated SOD1 can occur independently of SOD1 mutations.

Response: *We have added the following text and added citations to this and other references:*

"An insufficient level of Cu for SOD1 maturation has been reported in bulk CNS tissues of the hSOD1^{G37R} mouse model, whereas Zn levels were elevated in line with protein overexpression⁴⁶. Evidence of Cu dyshomeostasis and wild-type SOD1 dysfunction has also been observed in regions of neuronal loss in cases of Parkinson's disease which is linked with oxidative stress.⁴⁷"

Reviewer #2: An additional paper from the same team as above examined human spinal cord from

cases of ALS (PMID: 35512359). This paper should also be cited due to evidence provided for the presence of SOD1 in the human tissue with a compromised metalation state.

Response: *The following text and citation has been added:*

“It is also possible that Zn-deficient hSOD1^{G93A} species are present, as reported in human spinal cord for various cases of ALS, which would produce a perturbed higher-order structure.⁵¹”

Reviewer #2: The methods section needs to describe how motor neurons were identified relative to other Nissl-positive structures.

Response: *Motor neuron identification criteria are now described in the Methods.*

--

Reviewer #3: The method of nanoDESI-MS imaging for protein MS imaging is not new, several previously published papers have demonstrated the capability, with a few listed below:

J. Am. Chem. Soc. 2022, 144, 5, 2120–2128 (from the authors' group);

Angew. Chem., Int. Ed. 2022, 61, e202200721 (Julia Laskin group).

Response: *It is true that we have previously demonstrated native nano-DESI imaging of proteins and the manuscript cites our previous work (including the JACS reference mentioned). The paper from the Laskin group does not describe native mass spectrometry imaging of proteins, instead the focus is on analysis of unfolded proteins by use of denaturing solvents, i.e., any protein complexes are dissociated.*

Reviewer #3: 1. Proper control experiments are needed to claim the native metal binding is not affected by the existence of high concentrations of other prevalent metal cations (Na⁺, K⁺, NH₄⁺, etc.) Could test native metal-binding protein by spiking different salts. Although the authors indicated that m/z values of the detected MS1 peaks (e.g., Fig S7) matched with different binding numbers of Cu/Zn to the dimers. It is also very likely some or even a large portion of these comes from 3n*Na⁺ adduction as Na⁺ is prevalent in biological tissues. E.g., the mass increment of [M+3*Na+n*H] to [M+(n+3)*H] is 66 Da, very close to the addition of Cu or Zn, of which the isotopic mass is 63~68 Da. And this is very likely to be true if looking at all listed HCD MS2 spectra (Fig. S8-S10). From zoom-in spectra of the monomeric fragment ions, 2 small peaks almost equally distributed (~22 Da mass difference) between 2 adjacent monomers indicated different metalated states could be clearly seen, which may correspond to +1*Na⁺ adduction and +2*Na⁺ adduction. If the metalation is only Cu/Zn, the monomeric products dissociated from dimers should only have mass differences over 60 Da and those peaks with 22 Da mass differences should not be observed. Please comment on the formation of these small peaks in HCD MS2 spectra.

Response: *Mass spectrometer source optics were optimised for adduct and solvent removal. High resolution (240,00 FWHM at m/z 200) SIM mass spectra obtained from G93A brain confirm that these peaks are not the result of nNa⁺ adduction, see new Figure S14 , Supporting Information. We observe only two 6+ monomeric hSOD1^{G93A} complexes; with one and two metal ions (Zn or Cu). If the mass increment was due to adduction of Na⁺ ions we would first expect a series of signals starting at the apoSOD1G93A monomer and incrementing by ~22 Da, as demonstrated in the simulated mass spectrum (Figure S14 (bottom)). We do not detect the apo monomer and Na⁺ adduct series in the*

SIM spectra. Furthermore, we observe that the calculated peaks for the 3Na⁺ adduct do not align with the peaks in the SIM spectrum. Other adduct series (e.g. K⁺) are also not detected. Lastly, if salt adduction were occurring, we would expect other protein signals to be affected and this is not the case.

With respect to the HCD MS2 spectra, the peaks described by the reviewer are attributable to common neutral loss product ions resulting from the collisional activation of the dimer at m/z 3197¹⁰⁺, as shown in new Figure S11b, Supporting Information. Each major metal bound product ion features a loss of ~18 Da, i.e., H₂O. Other peaks are further neutral losses, e.g., CO₂ ~44 Da.

Reviewer #3: 2. The authors indicated that “Ion images for monomeric hSOD1wt and hSOD1G93A showed similar spatial distributions to the dimers of equivalent metal-binding state” and concluded that “demetalation is a key pathological change observed for SOD1”. These still need more critical examination. It is important to diminish overlapping intensities of dimers at 2x charge state as monomers. It is good that the authors noticed it and attempted to omit 10+ dimers and 5+ monomers and only used 11+ and 9+ for dimers and 6+ for monomers. However, from Fig. S7, 10+ is actually the most abundant charge state for dimers. 6+ may not be determined as the most abundant charge state for monomers and 5+ could be very likely the most. This elimination would largely obscure the quantitative accuracy when dimer/monomer ratios are investigated as they are comparing things not at the same criteria, integrating only 1 charge state for monomer and 2 charge states for dimers.

Response: *We have performed additional high mass resolution mass spectrometry in regions of high SOD1 metal deficiency (e.g., XII) from which we calculate that approximately 52% of the signal for 10+/5+ peak is contributed by dimer species overlap. These new data are included in Figure S31, Supporting Information. Based on this approximation, we have included the overlapping signals in the assessment of relative abundance of SOD1 complexes (Figs 2o and 3o).*

Reviewer #3: Besides, 6+ monomer still overlaps with 12+ dimer. How to make ratio calculations as monomer intensity cannot be obtained reliably? This also places concerns about the MS images of monomers. The co-localization of monomer (sum peak intensity at 6+ charge state only) could be false-positive and be attributed to the presence of dimer at the same m/z (at 12+ charge state).

Response: *Dimers in the 12+ charge state were not detected under our experimental conditions. Confirmation of their absence was confirmed by proton transfer charge reduction (PTCR) MS, now shown in Figure S12, Supporting Information. If the 6+ monomer overlapped with the 12+ dimer, we would expect to see a product ion series corresponding to the dimer in odd charge states; however, these signals were not detected. We are therefore confident that the monomer 6+ charge state is the sole contributor to this peak.*

Reviewer #3: 3. Furthermore, there are also concerns about the proportion of in-source dissociation of dimers to monomers and if that would be influential to the MS imaging of monomers. From the manuscript, such evaluation has not been done in their nanoDESI-MS setup. This could be done with protein standards. Can the authors provide more results and comments on this aspect?

Response: *Please see also response to Reviewer 1 above. We have added evaluation of source dissociation by comparison of a SIM spectrum and an HCD spectrum (both use identical source*

settings used in the imaging experiments: source dissociation 80 V, RF lens amplitude 120%). When SOD1 dimers dissociate by collisional activation, the endogenous metal ions rearrange to produce non-endogenous complexes i.e. 3 metal bound monomers. We found no evidence of 3 metal bound monomers in the SIM spectra. We also don't find evidence of complete metal ion dissociation (i.e. apo-SOD1), nor neutral loss product ions (e.g. H₂O). Conversely, during targeted collisional activation by HCD, monomers (3 metal ions, 0 metal ions) and neutral loss products (-H₂O, -CO₂) were detected. Thus we determined the source settings used for the SIM method to be suitably gentle to avoid in-source dimer → monomer dissociation.

Reviewer #3: 4. This work does not have sufficient evidence to support the conclusion that only demetalation, not monomerization, correlates with SOD1-G93A transgenic mouse pathology.

Response: As described above, we have performed additional mass spectrometry experiments on spinal cord tissue, and have performed histology and neuron counting experiments in both spinal cord and brain. Altogether, these data provide evidence that pathology correlates with metal-deficient SOD1 and not monomers.

Reviewer #3: 5. Based on the mass spectra shown in Figure S7, the signal intensities of hSOD1(s) were generally much higher in the G93A mouse brain, and the signal-to-noise was very poor in the WT mouse.

Response: The reviewer is correct. hSOD1G93A is expressed more highly than the wild-type levels, so the poorer signal intensity for hSOD1wt was expected. Nevertheless, it is comparable to the endogenous CAH signal (Figure S3 and S4) and sufficient for our analysis.

Reviewer #3: It looks like the results shown in Figure 2 were based on Figure S7. The authors need to better demonstrate the reproducibility of the method, and address the variation issue. Some additional data about the reproducibility and stability of the method would help to address these concerns.

Response: Our previous publications, in which the technique of native ambient mass spectrometry imaging was introduced, addressed the issue of reproducibility of the method (see <https://pubs.acs.org/doi/10.1021/acs.analchem.0c05277> and <https://onlinelibrary.wiley.com/doi/full/10.1002/anie.202201458>).

In this work, two biological replicates of the hSOD1G93A and hSOD1wt spinal cord samples (totaling 4 VH and 4 DH for each genotype), and three biological replicates of hSOD1G93A and hSOD1wt brains were analysed, as outlined in Table S1. In addition to images in the main figures, replicate images are shown in Figures S15, S16 & S167 (spinal cord), Figures S23, S24, S25, S27 and S30 (brain), Supporting Information.

We have added additional text to the Methods section describing the stability metrics:

“Electrospray stability was assessed using the ion trap mass analyser by monitoring m/z 307.21 until an RSD% < 15% was achieved by tuning flow rate, electrospray voltage and emitter position.”

and

“Proceeding with imaging experiments was conditional on achieving a signal intensity $>1 \times 10^4$ (normalized level) for holo-hSOD1^{G93A} dimer (10+ charge state) under these instrument conditions.”

Reviewer #3: 6. I would suggest the authors to add some protein standards in the DESI solvent to nominalize the data detected from the WT and G93A mice brain.

Response: *This is a good suggestion for the development of the NAMS technique for protein imaging, but normalisation to an in-solution protein standard is non-trivial and warrants a dedicated separate investigation that is beyond the scope of this manuscript.*

Reviewer #3: 7. Figure S14 d-f, higher signals of hSOD1 were detected from the area off the tissue. This observation showed that there is severe chemical delocalization for the tissue being imaged, further casting doubt of the conclusions drawn from this data.

Response: *On inspection, we agree that there was evidence of delocalisation in this experiment, which corresponded to a degraded tissue edge. The analysis for this brain was repeated on an alternative tissue section and these data are now included in place of the original.*

Reviewer #3: 8. On page 10, since the Zn-binding might inhibit Cu-ion incorporation and future maturation, the authors are expected to highlight the Zinc and Copper binding separately in the deconvolution progress. Specifically, do holo-monomers consist of the same ratio of Zn and Cu as the holo-dimers? If there is an unstated difference in preference monomerization towards Zn²⁺, then we cannot reach this conclusion by merely comparing the intensity of the monomer peak. This is highly confusing as Supplementary figures 7-11 & 17 are all annotated as “Zn/Cu ions”. In comparison, there is another article published in 2022 (<https://doi.org/10.1093/brain/awac165>), on page 3119, the author specifically clarified that “Zinc-deficient mature SOD1 is unlikely to undergo self-assembly from a native-like dimeric state”, and in this in-vivo testing, the Cu:Zn ratio within SOD1 aggregate is carefully measured.

Response: *We believe the labelling of “Zn/Cu” ions in the spectra legends may have caused confusion. We do not address a Zn:Cu ratio, rather the labelling was intended to show that the metals bound could be either Zn or Cu. We are currently unable to unambiguously distinguish between Cu and Zn ions bound to SOD1, only count the number of metal ions.*

The legend on each spectrum has been amended for clarity.

Reviewer #3: 9. For the PTM analysis, glutathionylation at Cys111 shown in article is previously found strongly associated with structural stability. But there are also many other phosphorylation, acetylation, and ubiquitination sites (<https://doi.org/10.1016/j.redox.2019.101270>). The authors are expected to examine how much extent the other PTMs may have an effect on unfolding.

Response: *All hSOD identified featured N-terminal acetylation, including S-glutathionylated hSOD^{G93A}. We also found that some proportion of the hSODG93A dimer (2 metal ions) has the disulfide bond intact (Figure S10), although a 2 Da mass difference is not currently resolvable when imaging. We have not identified other PTMs, likely because their abundance was below the limit of detection.*

Reviewers' Comments:

Reviewer #1:

Remarks to the Author:

The paper has been carefully edited and all of the comments from the reviewers have been addressed.

A point to consider in the future: The three metal dimer is likely to be the toxic species in vivo. With the current resolution of the methods, it is not proven that this species contains two zincs and one copper. However the heterodimer of Cu,Zn and zinc-deficient SOD is more toxic than the dimer of Cu-containing, zinc-deficient SOD. The cited Trist Brain paper establishes zinc-deficient SOD is present in human ALS tissues. The papers below show that this species is sufficient to induce the death of motor neurons.

Cu,Zn-superoxide dismutase increases toxicity of mutant and zinc-deficient superoxide dismutase by enhancing protein stability.

Garner MA, Ricart KC, Roberts BR, Bomben VC, Basso M, Ye Y, Sahawneh J, Franco MC, Beckman JS, Estévez AG.

J Biol Chem. 2010 Oct 29;285(44):33885-97. doi: 10.1074/jbc.M110.118901. Epub 2010 Jul 27.

PMID: 20663894 Free PMC article.

Induction of nitric oxide-dependent apoptosis in motor neurons by zinc-deficient superoxide dismutase.

Estévez AG, Crow JP, Sampson JB, Reiter C, Zhuang Y, Richardson GJ, Tarpey MM, Barbeito L, Beckman JS.

Science. 1999 Dec 24;286(5449):2498-500. doi: 10.1126/science.286.5449.2498.

PMID: 10617463

Reviewer #2:

Remarks to the Author:

Hale and colleagues have addressed the recommendations I made for revision. Making the connection to motor neuron numbers strengthens the study.

Reviewer #3:

Remarks to the Author:

The authors made proper revision and addressed most of my previous concerns. I think the revised manuscript is acceptable for publication.